# Genetic Variants of *HNF4A*, *WFS1*, *DUSP9*, *FTO*, and *ZFAND6* Genes Are Associated with Prediabetes Susceptibility and Inflammatory Markers in the Saudi Arabian Population

**DOI:** 10.3390/genes14030536

**Published:** 2023-02-21

**Authors:** Dalal N. Binjawhar, Mohammed G. A. Ansari, Shaun Sabico, Syed Danish Hussain, Amal M. Alenad, Majed S. Alokail, Abeer A. Al-Masri, Nasser M. Al-Daghri

**Affiliations:** 1Department of Chemistry, College of Science, Princess Nourah bint Abdulrahman University, Riyadh 11564, Saudi Arabia; dnbanjawhar@pnu.edu.sa; 2Department of Biochemistry College of Science, King Saud University, Riyadh 11451, Saudi Arabia; mansari@ksu.edu.sa (M.G.A.A.); ssabico@ksu.edu.sa (S.S.); shussain@ksu.edu.sa (S.D.H.); aalenad@ksu.edu.sa (A.M.A.); malokail@ksu.edu.sa (M.S.A.); 3Department of Physiology, College Medicine, King Saud University, Riyadh 11451, Saudi Arabia; aelmasri@ksu.edu.sa

**Keywords:** single nucleotide polymorphism, Saudi Arabia, genes related to insulin resistance

## Abstract

Prediabetes is a reversible, intermediate stage of type 2 diabetes mellitus (T2DM). Lifestyle changes that include healthy diet and exercise can substantially reduce progression to T2DM. The present study explored the association of 37 T2DM- and obesity-linked single nucleotide polymorphisms (SNPs) with prediabetes risk in a homogenous Saudi Arabian population. A total of 1129 Saudi adults [332 with prediabetes (29%) and 797 normoglycemic controls] were randomly selected and genotyped using the KASPar SNP genotyping method. Anthropometric and various serological parameters were measured following standard procedures. Heterozygous GA of *HNF4A*-rs4812829 (0.64; 95% CI 0.47–0.86; *p* < 0.01), heterozygous TC of *WFS1*-rs1801214 (0.60; 95% confidence interval (CI) 0.44–0.80; *p* < 0.01), heterozygous GA of *DUSP9*-rs5945326 (0.60; 95% CI 0.39–0.92; *p* = 0.01), heterozygous GA of *ZFAND6*-rs11634397 (0.75; 95% CI 0.56–1.01; *p* = 0.05), and homozygous AA of *FTO*-rs11642841 (1.50; 95% CI 0.8–1.45; *p* = 0.03) were significantly associated with prediabetes, independent of age and body mass index (BMI). Additionally, C-reactive protein (CRP) levels in rs11634397 (AA) with a median of 5389.0 (2767.4–7412.8) were significantly higher than in the heterozygous GA genotype with a median of 1736.3 (1024.4–4452.0) (*p* < 0.01). In conclusion, only five of the 37 genetic variants previously linked to T2DM and obesity in the Saudi Arabian population [*HNF4A*-rs4812829, *WFS1*-rs1801214, *DUSP9*-rs5945326, *ZFAND6*-rs11634397, *FTO*-rs11642841] were associated with prediabetes susceptibility. Prospective studies are needed to confirm the potential clinical value of the studied genetic variants of interest.

## 1. Introduction

Type 2 diabetes mellitus (T2DM) is one of the most prevalent multifactorial chronic disorders, characterized by impaired glucose tolerance and insulin sensitivity, leading to hyperglycemia in both fasting and postprandial states [1,2,3]. It is a rising epidemic of the last century, globally affecting 536.6 million adults aged 20–79 and projected to escalate to 783.2 million by 2045 [4]. The Kingdom of Saudi Arabia (KSA) is not immune to this global epidemic, with an estimated 7 million people in KSA affected by type 2 diabetes mellitus (T2DM) and 3 million with prediabetes [5]. The International Diabetes Federation (IDF) forecasted that the Middle East region would experience the most significant relative growth in the T2DM population, whereas, according to the World Health Organization (WHO), KSA has the 2nd highest T2DM prevalence in the Middle East and 7th highest worldwide [4,5]. This rise in T2DM prevalence is alarming, as it is associated with or triggers various chronic, acute, macro-, and microvascular complications [6,7,8], which significantly impact the quality of life and exert a socioeconomic burden. The estimated global economic burden was 966 billion USD in 2021, which is estimated to possibly escalate to 1054 billion USD by 2045 [4].

Prediabetes is an intermediate stage of T2DM characterized by above normal blood glucose or HbA1c levels, but not high enough to meet the diabetes threshold [9]. It has been linked to a higher risk of chronic diabetes-related complications [10,11]. The rapid industrialization and urbanization in KSA resulted in a notable rise in living standards, leading to a Westernized lifestyle where unhealthy food patterns and limited physical activity dominate. Moreover, age, obesity, and a sedentary lifestyle are the conventional risk factors for T2DM [8,12,13]. Additionally, the Saudi population appears to have a genetic predisposition to T2DM, which might be due to the high prevalence of consanguineous marriages and gestational diabetes mellitus (GDM) [14,15]. Screening plays a significant role, as prediabetes is amenable to interventions that prevent/delay the transition to overt T2DM and reduce the risk of T2DM-related complications [16,17,18]. Aside from screening, a multidimensional approach is desperately needed to holistically understand this disease. Studying gene variants that affect DM phenotypes might be an effective tool in predicting and preventing prediabetes and related complications.

In a milestone meta-analysis review involving 21 genome-wide association studies covering almost 123,000 individuals, with a replication set involving another 76,000 individuals, several genetic loci were identified with direct associations to glycemic traits, glucose homeostasis, and insulin resistance [19]. These novel loci were successfully replicated in a homogenous Chinese population, thus reinforcing their impact on diabetes risk [20]. 4Consequently, we previously assessed the association of these genetic loci with T2DM risk based on identified genetic variants that increase T2DM susceptibility in other populations and found that nine variants [*WFS1*, *JAZF1*, *SLC30A8*, *CDKN2A/B*, *TCF7L2*, *KCNQ1*, *HMG20A*, *HNF4A*, and *DUSP9*] were associated with T2DM in the Saudi population [21]. In addition, the same previously genotyped cohort was also investigated for its possible link to obesity, revealing only five allelic variants [*FTO* and *TCF7L2* genes] out of 37 were associated with obesity in the Saudi Arabian population [22]. Given this information, it makes sense to further examine the individual and cumulative influences of these genetic variants on the development and susceptibility to prediabetes, as this intermediate stage to T2DM has yet to be investigated in the Arabian population. Given that genetic testing and interventional gene therapy is rapidly taking shape as an emerging field in medicine, genetic variant studies are needed to fully understand the molecular basis of common human disorders, including insulin-resistant diseases and, in this, case, intermediate stages (prediabetes) of harder outcomes (T2DM). Therefore, the rationale of the current study was to explore the association of previously established 37 SNPs with prediabetes susceptibility in the Saudi Arabian population.

## 2. Materials and Methods

### 2.1. Study Design and Population

In this cross-sectional study, 1129 Saudi adults (332 with prediabetes and 797 with normoglycemia) aged between 30 and 60 years were randomly selected from the Biomarker screening project in Riyadh (RIYADH COHORT) [21]. This is a capital-wide epidemiological study comprising over 17,000 consenting Saudis recruited from various Primary Health Care Centers (PHCCs) in Riyadh, KSA. Demographic data and medical history were obtained through a self-administered general questionnaire. Moreover, written informed consent was obtained from all participants before inclusion in the study. Subjects taking anti-diabetic drugs or any medication known to affect glucose homeostasis were excluded from this study. Additionally, subjects who were morbidly obese, had thyroid disorders, including history of hyperparathyroidism, hypercalcemia, chronic kidney disease, or significantly affected with comorbidities that would interfere with study participation were excluded from the study. The study was conducted in accordance with the Declaration of Helsinki. Permission to collect samples from the different PHCCs were provided by the Ministry of Health, General Directorate of Affairs in Riyadh, KSA (No. 74191, dated Hijri 25/05/1434, corresponding to 13 April 2013).

### 2.2. Biochemical Analysis

Participants were requested to report to their allocated PHCCs after an overnight fast (>10 h) for anthropometric analysis and blood sampling. Peripheral blood was obtained in EDTA tubes for DNA extraction, while plain tubes were used to collect blood for serum analysis. Extracted serum and EDTA tubes were transferred to the Chair for Biomarkers of Chronic Diseases (CBCD) laboratory and stored at −20 °C until further analysis. Various anthropometric parameters were recorded, as mentioned in our previous study [23]. Fasting glucose and lipid profile (high- and low-density lipoprotein cholesterol, total cholesterol, and triglycerides) were measured routinely using an autoanalyzer (Konelab, Vantaa, Finland) [24]. Pro-inflammatory cytokines, including tumor necrosis factor α (TNF-α) and interleukins IL6 and IL1β, were measured using commercially available multiplex immunoassay kits that utilize the Luminex xMAP technology platform (Luminex Corporation, Austin, TX, USA), which enables simultaneous analysis of multiple biomarkers in human serum. For TNFα: intra-assay <10% coefficient of variation (CV), inter-assay <20% CV. For IL-6: intra-assay <10% CV, inter-assay <15% CV. For IL-1β: intra-assay <10% CV, inter-assay <15% CV. C-Reactive protein levels were quantified using a commercial enzyme-linked immunosorbent assay (ELISA) kit (Human C-Reactive Protein/CRP Quantikine ELISA Kit, R&D systems, Minneapolis, MN, USA) following the manufacturer’s instructions.

### 2.3. Prediabetes Screening

The operational definition of prediabetes used in the present study was based on the cut-off provided by the American Diabetes Association (ADA), which is a fasting blood glucose level of 5.6–6.9 mmol/L (100–125 mg/dL) [25]. The fasting blood glucose level was measured using an automatic biochemical analyzer. For the purpose of this study, the fasting blood glucose was preferred over a 2 h oral glucose tolerance test (for the diagnosis of impaired glucose tolerance) since it is more practical for large-scale screening.

### 2.4. Genotyping

Genomic DNA was extracted from the blood using the Blood Genomic Prep Mini Spin Kit (GE Healthcare, Chicago, IL, USA). A Nanodrop spectrophotometer (ND-1000, NanoDrop Technologies by Thermo Fisher Scientific, Wilmington, DE, USA) was used to quantify the concentrations of purified DNA (260/280). The 37 SNPs (rs7903146, rs5015480, rs12779790, rs10923931, rs10440833, rs11899863, rs13081389, rs3802177, rs849134, rs5215, rs1470579, rs6795735, rs1387153, rs243021, rs7578326, rs4457053, rs972283, rs896854, rs13292136, rs2311362, rs1552224, rs7957197, rs11634397, rs8042680, rs5945326, rs163184, rs4430796, rs4812829, rs1802295, rs7178572, rs2028299, rs3923113, rs16861329, rs1531343, rs1801214, rs10965250, and rs11642841) were evaluated in prediabetes subjects and their normoglycemic counterparts using the KASPar method (KbioScience, Hoddesdon, UK), with a genotype success rate of 99.1% according to our earlier described work [21].

### 2.5. Statistical Analysis

Data was analyzed using SPSS version 21.0 software. Categorical variables were presented as N (%). Hardy–Weinberg (HW) distribution was assessed for the genotypes in the prediabetes group and their healthy counterparts. Normal quantitative variables were presented as mean (SD) and non-normal quantitative variables were presented as median (quartile 1–quartile 3). The independent samples t-test and Mann-Whitney U-test were used to determine statistical differences between normal and prediabetes subjects for normal and non-normal quantitative variables, respectively. The Kruskal–Wallis test was used to determine statistical differences between SNPs for respective quantitative variables. Bonferroni corrections were used to adjust for multiple comparison. Logistics regression was used to determine the association between prediabetes and SNPs. Furthermore, the effects of covariates including age, gender, and BMI, were removed to obtain the adjusted odds ratios (OR) with 95% confidence interval (CI). A *p* value < 0.05 was considered statistically significant.

## 3. Results

### 3.1. General Characteristics

The anthropometric, clinical, and biochemical characteristics of the studied population according to prediabetes status are shown in Table 1**.** The prevalence of prediabetes in the studied population was 29.4%. The prediabetes group was significantly older than the control group (*p* < 0.01) and while the percentage of males was higher in the prediabetes group (45%) than in the control group (38%) (*p* = 0.03), prediabetes was more common in women than men. Measurements of weight, BMI, waist, hips, waist-hip ratio (WHR), fasting glucose, and triglycerides were significantly higher in prediabetes subjects than in controls (*p* < 0.01). No significant differences were seen between the prediabetes and control groups in terms of pro-inflammatory cytokines measured.

### 3.2. Association of T2DM-Related Genetic Variants with the Occurrence of Prediabetes

A logistic regression analysis of the genotypes with the five SNPs is presented in Table 2**.** Prediabetes risk increased by 57% among participants with the homozygous AA genotype of rs11642841 (*FTO*) compared to the CC genotype (*p* = 0.02). After adjusting for age, gender, and BMI, the risk was reduced to 50%. Furthermore, heterozygous GA of rs4812829 (*HNF4A*), rs5945326 (*DUSP9*), and rs11634397 (*ZFAND6*), along with heterozygous TC of rs1801214 (*WFS1*), were associated with a decreased risk for prediabetes.

The relationship between the 37 T2DM-related SNP loci and predisposition to prediabetes was assessed by applying a logistic regression model using age, gender, and BMI as covariates. Among the 37 SNPs, five SNPs, including *FTO* (rs11642841), *HNF4A* (rs4812829), *WFS1* (rs1801214), *DUSP9* (rs5945326), and *ZFAND6* (rs11634397), showed significant associations with prediabetes (*p*-values = 0.03, <0.01, <0.01, 0.01, 0.05, respectively) (Appendix A).

Genotype frequencies of all the significant SNPs (rs11642841, rs4812829, rs1801214, and rs11634397) did not deviate from Hardy-Weinberg equilibrium in our population except for rs5945326 **(**Appendix A).

### 3.3. Association of Five Selected Genetic Variants with Anthropometric Measures

Table 3 shows the median and quartiles of anthropometric data according to the studied polymorphisms. The median and quartiles of weight (*p* = 0.04) and BMI (*p* = 0.02) were significantly higher in the AA genotype than in the CC genotype of rs11642841 (*p* < 0.05). In addition, the median and quartiles of WHR in the GG genotype were higher than in the GA and AA genotypes of rs5945326 (*p* < 0.01).

### 3.4. Association of Five Selected Genetic Variants with Inflammatory Markers

We assessed the association of the five selected SNPs with various inflammatory markers (Table 4). In rs11634397, CRP levels of the homozygous genotype (AA) with a median of 5389.0 (2767.4–7412.8) were significantly higher than those of the heterozygous GA genotype with a median of 1736.3 (1024.4–4452.0) (*p* < 0.01). Additionally, TNF-α, CRP, and IL-1β levels were associated with rs11634397, rs4812829, and rs1801214, respectively. This significance was lost in post-hoc analysis.

## 4. Discussion

Epidemiological data reveal that approximately 5–10% of prediabetes subjects will develop diabetes each year and an equal percentage will return to normal [9]. In the past three decades, reports from KSA suggest a ten-fold rise in diabetes prevalence, which is anticipated to rise globally [5]. Identifying genetic markers potentially enables early detection and reduces the risk of T2DM prognosis and related complications. This study was primarily aimed at exploring the association of 37 T2DM-related genetic variants with prediabetes. These variants of interest conferred susceptibility to T2DM in European and South Asian diabetes populations [26,27] and were subsequently replicated in Saudi Arabian ethnic groups [21,22]. The current study revealed that five out of 37 genetic variants were associated with prediabetes and inflammation among Saudi Arabian adults. Interesting to note was the high prevalence of prediabetes (29%) in the group and the anticipated worse cardiometabolic profile of individuals with prediabetes compared to controls, including being significantly older and the substantially higher prevalence in women.

Hepatocyte nuclear factor 4-α (*HNF4A*) regulates hepatic gluconeogenesis and insulin secretion. It belongs to the nuclear receptor superfamily and plays a crucial role in glucose homeostasis in pancreatic β cells and the liver [28,29]. The corresponding gene is located on chromosome 20q13 and is directly implicated in insulin gene expression [28]. The present study is the first to report an association between the *HNF4A* gene variant (rs4812829) with prediabetes, suggesting that the heterozygous genotype (GA) is protective of prediabetes risk in Saudi Arabian adults. However, Wang et al. tested and linked P2 promoter polymorphism rs1884613 of *HNF4A* with prediabetes susceptibility in the Chinese Han population [30]. In other populations, a genome-wide association (GWA) study reported that the risk allele of rs4812829 was significantly associated with T2DM and (GDM) risk in a South Asian cohort [31,32]. Conversely, rs4812829 has also been associated with obesity [33,34]. Moreover, multiple studies in different ethnicities have linked T2DM susceptibility with *HNF4A* variants [29,31,35,36,37]. Interestingly, it was shown that *HNF4A* variants play a role in type I maturity-onset diabetes of the young (MODY) by impairing insulin sensitivity and β-cell function [38].

*WFS1* encodes several proteins, including Wolframin, which is embedded in the endoplasmic reticulum membrane. It is widely expressed across various organs, particularly in the brain and pancreas [39]. Several studies have linked variations in the *WFS1* gene to Wolfram Syndrome, an autosomal recessive disorder, and T2DM susceptibility [40,41,42,43]. In mice, *WFS1* disruption resulted in increased glucose intolerance and insulin deficiency [44]. However, the underlying effect of these variants on the prediabetes phenotype has not been explored much. In the current study, heterozygous TC of the *WFS1*-rs1801214 variant located in the coding sequence showed a statistically significant association with prediabetes (0.60; 95% CI 0.44–0.80; *p* < 0.01). To the best of our knowledge, no studies have shown a relationship between the *WFS1*-rs1801214 variant and prediabetes risk; however, Sparsø et al. revealed the interplay between other variants in *WFS1* (rs734312, rs10010131) and the prediabetes phenotype [45].

*DUSP9* and *ZFAND6* are expressed in various tissues with a significant role in glucose homeostasis. The current study revealed that their variants, including the GA heterozygous of *DUSP9* (rs5945326) and GG heterozygous of *ZFAND6* (rs11634397) genotypes, contributed to decreased risk of developing prediabetes. The GA heterozygous of rs5945326 genotype was also associated with higher waist measurement and WHR. Moreover, the AA homozygous genotype had significantly higher CRP levels than the GA heterozygous genotype. These gene loci were associated with T2DM and β-cell dysfunction and are believed to play a prominent role in protecting against versus developing insulin resistance. There was no prior knowledge available revealing the impact of the various genotypes studied in the current study, including heterozygous GA of *HNF4A*-rs4812829, (0.64; 95% CI 0.47–0.86; *p* < 0.01), heterozygous TC of *WFS1*-rs1801214 (0.60; 95% CI 0.44–0.80; *p* < 0.01), heterozygous GA of *DUSP9*-rs5945326 (0.60; 95% CI 0.39–0.92; *p* = 0.01), heterozygous GA of *ZFAND6*-rs11634397 (0.75; 95% CI 0.56–1.01; *p* = 0.05), and homozygous AA of *FTO*-rs11642841, on prediabetes susceptibility.

*FTO* located on 16q12.2 is substantially associated with elevated basal metabolic rate and T2DM [46]. Numerous *FTO* gene variants have been reported with an amplified contributory effect on T2DM and obesity [47,48]. However, our study showed that SNP rs1164284 had the most substantial susceptibility to prediabetes in the Arab population. Furthermore, among obesity-related traits, weight, BMI, and waist were recognized to be the most significantly associated with the homozygous genotype AA compared to the CC genotype. Importantly, findings from our previous study and other research are concurrent and support the current study outcomes, which suggests that BMI and waist circumference could be potential T2DM predictors. Among the various anthropometric parameters, WC was found to be a significant early marker of T2DM [22,46,49].

Several variants in the same or different genes act synergistically and affect diabetes phenotypes. Initially, the association of 37 T2DM-related SNPs with T2DM was reported in the European population. Subsequently, we replicated and identified around nine loci, with a significant effect on the development of T2DM in the Saudi Arabian population [21]. However, no study has shown the role/relationship of these SNPs in the development of prediabetes. For the first time, we found five loci with an independent effect on prediabetes susceptibility. It is not surprising that the risk alleles of these SNPs are all associated with pancreatic β-cell dysfunction. Even though the significance of prediabetes has been highly underscored, very few studies have assessed the diabetogenic impact of genetic variants.

## 5. Limitation

The authors acknowledge certain limitations. Fasting glucose instead of a 2 h glucose tolerance test was used for prediabetes screening; hence, there is a risk of categorizing individuals with impaired glucose tolerance under the normoglycemic group. Nevertheless, the use of fasting glucose was justified as it is more practical for screening large numbers of participants. Additionally, the subjects of the current study are of Saudi Arabian ethnicity; hence, the outcomes of this research might not apply to other populations. However, the study’s strength includes the homogeneity of the population, providing first-hand evidence of the association of the studied SNPs with prediabetes in individuals of a homogenous Arab ethnic group.

## 6. Conclusions

In summary, we found significant associations between prediabetes risk and five variants closely related to the *FTO*, *HNF4A*, *WFS1*, *DUSP9,* and *ZFAND6* genes (rs4812829, rs1801214, rs5945326, rs11642841, and rs11634397) among Saudi Arabian adults. Prospective studies involving metabolically healthy individuals should be conducted to assess the true value of the investigated polymorphisms as risk factors for prediabetes and T2DM.

## Figures and Tables

**Table 1 genes-14-00536-t001:** Anthropometry and Clinical Characteristics of Subjects.

Parameters	Control	Prediabetes	*p*-Values
N	797 (70.6)	332 (29.4)	
Age (years)	41.5 ± 12.6	45.7 ± 13.9	<0.01
Male/Female	301/496	149/183	0.03
Weight (kg)	72.0 (62.0–83.0)	77.0 (67.0–88.0)	<0.01
BMI (kg/m^2^)	27.5 (24.3–31.6)	29.0 (25.4–33.8)	<0.01
WHR	0.88 (0.81–0.94)	0.89 (0.84–0.97)	<0.01
Waist (cm)	90.0 (79.0–101.0)	95.0 (83.0–104.0)	<0.01
Hip (cm)	103.0 (94.0–112.0)	106.0 (95.0–115.0)	0.011
HDL-cholesterol (mmol/L)	0.9 ± 0.4	0.9 ± 0.3	0.11
LDL-cholesterol (mmol/L)	3.4 ± 1.0	3.4 ± 1.0	0.65
Total cholesterol (mmol/L)	5.0 ± 1.0	5.1 ± 1.0	0.47
Triglycerides (mmol/L)	1.3 (1.0–1.9)	1.5 (1.1–2.0)	<0.01
Fasting Glucose (mmol/L)	4.9 (4.5–5.3)	5.9 (5.7–6.0)	<0.01
TNF-α (pg/mL)	5.1 (1.6–10.2)	5.7 (2.6–8.1)	0.89
CRP (ng/mL)	3147.2 (1250.8–5851.2)	2767.4 (1111.7–5780.0)	0.54
IL-6 (pg/mL)	3.7 (1.9–11.2)	2.5 (1.7–6.7)	0.48
IL-1β (pg/mL)	1.0 (0.8–1.2)	1.2 (0.6–2.8)	0.61

Note: Data presented as mean ± SD for normal variables and median (Q1–Q3) for non-normal variables. BMI, body mass index; WHR, waist-hip ratio; HDL, high-density lipoprotein; LDL, low-density lipoprotein; TNF-α, tumor necrosis factor α; CRP, C-reactive protein; IL-6, interleukin 6; IL-1β, interleukin 1β; *p* < 0.05 was considered significant.

**Table 2 genes-14-00536-t002:** Association between SNPs and prediabetes status.

SNPs		Control	Prediabetes	Unadjusted		Adjusted	
				OR (95%CI)	*p*-Value	OR (95%CI)	*p*-Value
rs11642841	CC	332 (42.2)	124 (37.6)	1		1	
	CA	349 (44.3)	144 (43.6)	1.10 (0.83–1.47)	0.50	1.08 (0.81–1.45)	0.58
	AA	106 (13.5)	62 (18.8)	1.57 (1.08–2.28)	0.02	1.50 (1.02–2.21)	0.03
rs4812829	GG	485 (61.6)	231 (70.0)	1		1	
	GA	273 (34.7)	83 (25.2)	0.64 (0.48–0.85)	<0.01	0.64 (0.47–0.86)	<0.01
	AA	29 (3.7)	16 (4.8)	1.16 (0.62–2.18)	0.65	1.17 (0.62–2.22)	0.63
rs1801214	TT	222 (28.4)	129 (39.0)	1		1	
	TC	405 (51.7)	142 (42.9)	0.60 (0.45–0.81)	<0.01	0.60 (0.44–0.80)	<0.01
	CC	156 (19.9)	60 (18.1)	0.66 (0.46–0.96)	0.03	0.67 (0.46–0.98)	0.04
rs5945326	AA	628 (79.5)	282 (85.2)	1		1	
	GA	122 (15.4)	30 (9.1)	0.55 (0.36–0.84)	<0.01	0.60 (0.39–0.92)	0.01
	GG	40 (5.1)	19 (5.7)	1.06 (0.60–1.86)	0.84	1.03 (0.58–1.83)	0.92
rs11634397	GG	244 (30.8)	114 (34.4)	1		1	
	GA	402 (50.8)	142 (42.9)	0.76 (0.56–1.01)	0.06	0.75 (0.56–1.01)	0.05
	AA	146 (18.4)	75 (22.7)	1.10 (0.77–1.57)	0.60	1.06 (0.74–1.53)	0.75

Note: Data presented as N (%) and odds ratio (95% CI) obtained from logistic regression; adjusted indicates results adjusted for covariates, i.e., age, gender, and BMI. *p*-value < 0.05 was considered significant.

**Table 3 genes-14-00536-t003:** Anthropometric data according to selected SNPs.

SNPs		Weight (Kg)	*p-*Value	BMI (kg/m^2^)	*p-*Value	Waist (cm)	*p-*Value	Hips (cm)	*p-*Value	WHR	*p-*Value
rs11634397	GG	73.0 (63.0–85.0)	0.10	27.8 (24.2–32.0)	0.65	90.0 (79.0–102.0)	0.46	102.0 (91.0–112.0)	0.16	0.9 (0.8–1.0)	0.11
	GA	73.5 (63.0–84.0)		27.8 (24.9–32.6)		90.5 (81.0–102.0)		104.0 (95.0–113.0)		0.9 (0.8–0.9)	
	AA	75.0 (62.5–84.0)		27.8 (24.3–33.2)		92.0 (82.0–102.0)		104.0 (94.0–114.0)		0.9 (0.8–0.9)	
rs5945326	AA	74.0 (64.0–85.0)	0.06	27.8 (24.6–32.5)	0.10	92.0 (81.0–102.0)	<0.01	104.0 (94.0–113.0)	0.57	0.9 (0.8–0.9)	<0.01 ^AB^
	GA	70.0 (60.0–82.0)		27.6 (24.2–32.9)		86.0 (74.0–98.3)		103.0 (94.0–113.0)		0.8 (0.8–0.9)	
	GG	76.5 (64.0–86.0)		28.0 (25.2–32.4)		96.0 (80.0–105.0)^B^		102.0 (90.0–113.0)		0.9 (0.9–1.0)	
rs4812829	GG	73.5 (63.0–84.5)	0.80	27.8 (24.8–32.7)	0.51	91.0 (80.0–102.0)	0.35	104.0 (95.0–113.0)	0.70	0.9 (0.8–0.9)	0.35
	GA	73.0 (62.0–84.0)		27.6 (24.2–32.2)		92.0 (81.0–103.0)		103.0 (94.0–113.0)		0.9 (0.8–0.9)	
	AA	76.5 (65.5–84.5)		28.2 (25.3–32.4)		88.0 (79.0–100.0)		103.0 (88.0–112.0)		0.9 (0.8–0.9)	
rs1801214	TT	71.3 (62.0–84.0)	0.38	27.4 (24.3–32.3)	0.40	91.0 (80.0–102.0)	0.53	102.0 (93.5–112.0)	0.48	0.9 (0.8–0.9)	0.60
	TC	74.0 (64.5–84.5)		28.0 (24.9–32.7)		92.0 (81.0–103.0)		104.0 (94.0–114.0)		0.9 (0.8–0.9)	
	CC	76.0 (64.0–85.0)		28.1 (24.6–32.4)		90.0 (80.0–100.5)		104.5 (95.0–113.0)		0.9 (0.8–0.9)	
rs11642841	CC	72.0 (63.0–84.0)	0.04	27.5 (24.2–31.8)	0.02 ^A^	89.5 (79.0–102.0)	0.06	102.0 (93.0–112.0)	0.08	0.9 (0.8–0.9)	0.29
	CA	73.0 (62.2–83.0)		27.7 (24.7–32.0)		91.0 (81.0–101.0)		104.0 (95.0–113.0)		0.9 (0.8–0.9)	
	AA	77.5 (65.0–88.0)^A^		29.8 (25.3–34.4)		95.0 (82.0–104.0)		106.5 (95.0–114.0)		0.9 (0.8–1.0)	

Note: Data presented as median (1st quartile–3rd quartile); *p*-values were obtained from Kruskal–Wallis test; *p* < 0.05 considered significant. Superscripts A and B indicate significance from 1st and 2nd genotypes, respectively.

**Table 4 genes-14-00536-t004:** Inflammatory markers according to selected SNPs.

SNPs		TNF-α (pg/mL)	*p-*Value	CRP (ng/mL)	*p-*Value	IL-6 (Pg/mL)	*p-*Value	IL-1β (Pg/mL)	*p-*Value
rs11634397	GG	6.4 (3.6–10.2)	0.05	3043.6 (832.8–7571.3)	<0.01	1.3 (0.7–3.7)	0.15	0.8 (0.7–2.0)	0.67
	GA	4.4 (1.3–8.1)		1736.3 (1024.4–4452.0)		4.8 (2.1–9.5)		1.2 (0.8–1.7)	
	AA	6.2 (2.2–12.0)		5389.0 (2767.4–7412.8) ^B^		5.6 (3.5–105.6)		0.8 (0.7–12.9)	
rs5945326	AA	6.1 (2.2–8.7)	0.10	2841.4 (1178.9–5780.0)	0.84	3.6 (1.7–7.0)	0.42	1.0 (0.7–1.7)	0.51
	GA	3.5 (1.1–4.7)		3019.2 (957.7–5780.0)		3.2 (0.9–22.4)		1.2 (1.0–1.5)	
	GG	6.5 (0.9–12.3)		4822.6 (2505.9–6129.5)		14.3 (14.3–14.3)		1.5 (0.8–2.2)	
rs4812829	GG	5.9 (2.1–9.9)	0.52	2841.4 (1286.8–5780.0)	0.05	5.2 (1.7–10.5)	0.88	1.2 (0.8–1.7)	0.84
	GA	5.4 (2.0–7.9)		3888.5 (1111.7–7884.9)		3.6 (1.7–7.7)		0.9 (0.6–3.0)	
	AA	2.7 (1.7–6.4)		140.9 (121.9–159.9)		2.7 (2.6–2.9)		1.2 (1.2–1.2)	
rs1801214	TT	6.1 (1.8–8.6)	0.15	3427.2 (1418.9–7412.8)	0.48	4.8 (2.2–6.3)	0.08	0.9 (0.8–1.2)	0.05
	TC	4.1 (1.4–8.4)		2615.3 (1024.4–5547.7)		2.9 (1.3–7.7)		1.0 (0.6–1.7)	
	CC	6.6 (4.2–8.7)		2869.1 (1250.8–5279.5)		105.6 (5.6–1493.2)		12.9 (1.2–98.9)	
rs11642841	CC	6.0 (1.3–9.1)	0.71	3748.1 (1500.0–5780.0)	0.54	3.7 (1.3–7.7)	0.58	1.0 (0.8–2.3)	0.93
	CA	5.3 (2.6–8.7)		2581.7 (991.1–5897.2)		2.9 (1.3–11.9)		1.2 (0.8–1.7)	
	AA	3.0 (1.8–7.9)		2767.4 (839.3–7412.8)		6.3 (2.1–12.9)		0.9 (0.7–1.8)	

Note: Data presented as median (1st quartile–3rd quartile); *p*-values were obtained from Kruskal–Wallis test; *p* < 0.05 was considered significant. Superscript B indicates significance from 2nd genotype.

## Data Availability

Data is available upon request to the corresponding author.

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
