# Peer review of "Genetic Variants of HNF4A, WFS1, DUSP9, FTO, and ZFAND6 Genes Are Associated with Prediabetes Susceptibility and Inflammatory Markers in the Saudi Arabian Population"

_genes, 2023, doi:10.3390/genes14030536_

Round 1

Reviewer 1 Report

Title: Genetic Variants of HNF4A, WFS1, DUSP9, FTO, and ZFAND6 genes are associated with Prediabetes Susceptibility and Inflammatory Markers in the Arab Population

Authors: Dalal N Binjawhar, Mohammed G.A.Ansari, Shaun Sabico, Syed Danish Hussain, and Nasser M. Al-Daghri

General comment:

In the world struggling with the obesity pandemic, we need reliable markers helpful to identify individuals prone to the metabolic complications of the disease. In their work, Dalal N Binjawhar et al investigated the association between the genetic polymorphism in genes related to carbohydrate metabolism with the risk of prediabetes development. The idea of the manuscript is appealing; however; there are several methodological aspects the Authors should consider.

Major revisions:

Introduction:

1)      Please justify briefly the selection of the investigated polymorphism as risk factors for prediabetes.

2)      “However, the prospective effect of these genetic variants on the development of Prediabetes has not been studied yet. Herein, the rationale of the current study is to explore the association of these previously established 37 SNPs and prediabetes susceptibility in the Arab population.” – since the study design is not longitudinal, it would be difficult to assess the prospective effect of the studied variants on the development of diabetes.

Materials and methods:

1)      Lines 80-82: “Additionally, subjects with a history of hyperparathyroidism, hypercalcemia, chronic kidney disease, or significantly affected with co-morbidities that would interfere with study participation were excluded from the study.” – please add information if thyroid disorders were also in the exclusion criteria.

2)      Lines 103-106: If the diagnosis of prediabetes was based on the fasting glucose measurement only, there is a risk that individuals with impaired glucose tolerance (another form of prediabetes) were included into a control group.

Results:

1)      Table 1 – please notice and comment that the prediabetes group was significantly older compared to the control group and the gender distribution differed significantly between the groups.

2)      Second Table 2 – please change the Table number to 3 (and subsequently Table 3 to Table 4) and consider rearranging this table (or use a horizontal page format) to make it more eligible.

Discussion:

1)      Please consider the limitations of the study resulting from its design. To assess the value of the investigated polymorphisms as a risk factors for diabetes, a prospective study of the group of metabolically healthy individuals should be performed.

2)      Please discuss the possible impact of age- and gender-related differences between the studied groups on the study's results.

3)      Please mention the methods of prediabetes diagnosis as a potential confounder.

Minor revisions:

1)      Whole manuscript – if the Authors decided to write the word "Prediabetes" with a capital letter, I would recommend doing it permanently.

2)      Lines 46-48: “This rise in DM prevalence is alarming as it is associated with or triggers various chronic, acute, Macro-and microvascular complications [6-8], which significantly impact the quality of life and exert a socioeconomic burden." – please remove the capital letter from "Macro."

3)      Lines 191-193: “This study is the first to report the association of the HNF4A gene variant (rs4812829) in the prediabetes population, which shows that the heterozygous genotype (GA) is significantly associated with a decreased risk of getting Prediabetes in the participants." – please consider to change the structure of this sentence.

4)      Lines 199-201: “Conversely, rs4812829 has also been associated with obesity {30, 31]. Moreover, multiple studies in different ethnicities have linked T2DM susceptibility with HNF4A variants [26,  28, 32-34].” – please change the reference format and the bracket.

5)      Lines 255-257: “In summary, we found the strongest associations between five variants located within or closely related to the FTO, HNF4A, WFS1, DUSP9, and ZFAND6 genes (rs4812829, rs1801214, rs5945326, rs11642841, and rs11634397).” – please write “and” without italics.

Author Response

Major revisions:

  1. Introduction: Please justify briefly the selection of the investigated polymorphism as risk factors for prediabetes.

Response: We thank the reviewer for this comment. Justification for the selection of investigated polymorphisms were added in the revised introduction.

  1. “However, the prospective effect of these genetic variants on the development of Prediabetes has not been studied yet. Herein, the rationale of the current study is to explore the association of these previously established 37 SNPs and prediabetes susceptibility in the Arab population.” – since the study design is not longitudinal, it would be difficult to assess the prospective effect of the studied variants on the development of diabetes.

Response: The comment is well received and amended accordingly.

  1. Materials and methods: Lines 80-82: “Additionally, subjects with a history of hyperparathyroidism, hypercalcemia, chronic kidney disease, or significantly affected with co-morbidities that would interfere with study participation were excluded from the study.” – please add information if thyroid disorders were also in the exclusion criteria.

Response: It has been added.

  1. Lines 103-106: If the diagnosis of prediabetes was based on the fasting glucose measurement only, there is a risk that individuals with impaired glucose tolerance (another form of prediabetes) were included into a control group.

Response: We thank the reviewer for this comment. Both IGT and IFG are considered intermediate stages of diabetes and are often used interchangeably even in the clinical practice here in Saudi Arabia. Given the large number of participants and practicality for use in epidemiology studies, fasting glucose was used instead 2-hour OGTT for the diagnosis of prediabetes. We nevertheless included this in the revised limitation section.

  1. Results: Table 1 – please notice and comment that the prediabetes group was significantly older compared to the control group and the gender distribution differed significantly between the groups.

Response: It has now been addressed in the revised results section.

  1. Second Table 2 – please change the Table number to 3 (and subsequently Table 3 to Table 4) and consider rearranging this table (or use a horizontal page format) to make it more eligible.

Response: We are thankful to the reviewer for highlighting it. Corrections have been made in the revised version.

  1. Discussion: Please consider the limitations of the study resulting from its design. To assess the value of the investigated polymorphisms as a risk factors for diabetes, a prospective study of the group of metabolically healthy individuals should be performed.

Response: We thank the reviewer for this comment. The limitation section has been expanded and the suggested statement was included in the revised conclusion section as it is more indicative of future investigations to strengthen the present findings.

  1. Please discuss the possible impact of age- and gender-related differences between the studied groups on the study's results.

Response: We thank the reviewer for this comment. Age- and sex-differences were mentioned in the revised first paragraph of the discussion section, which, as anticipated, explained the worse metabolic profile observed in the prediabetes group. Adjustments however are not done since the study involves genetic variants and its association with the disease of interest (prediabetes).

  1. Please mention the methods of prediabetes diagnosis as a potential confounder.

 Response: It has been mentioned in the revised limitation section.

Minor revisions:

  1. Whole manuscript – if the Authors decided to write the word "Prediabetes" with a capital letter, I would recommend doing it permanently.

Response: IT was revised accordingly. Prediabetes was used as a capital only if used as the beginning of the sentence.

  1. Lines 46-48: “This rise in DM prevalence is alarming as it is associated with or triggers various chronic, acute, Macro-and microvascular complications [6-8], which significantly impact the quality of life and exert a socioeconomic burden." – please remove the capital letter from "Macro."

Response: It has been revised accordingly.

  1. Lines 191-193: “This study is the first to report the association of the HNF4A gene variant (rs4812829) in the prediabetes population, which shows that the heterozygous genotype (GA) is significantly associated with a decreased risk of getting Prediabetes in the participants." – please consider to change the structure of this sentence.

Response: It has been revised accordingly.

  1. Lines 199-201: “Conversely, rs4812829 has also been associated with obesity {30, 31]. Moreover, multiple studies in different ethnicities have linked T2DM susceptibility with HNF4A variants [26, 28, 32-34].” – please change the reference format and the bracket.

Response: It has been changed accordingly.

  1. Lines 255-257: “In summary, we found the strongest associations between five variants located within or closely related to the FTO, HNF4A, WFS1, DUSP9, and ZFAND6genes (rs4812829, rs1801214, rs5945326, rs11642841, and rs11634397).” – please write “and” without italics.

Response: Italics were removed.

Reviewer 2 Report

The manuscript entitled {Genetic Variants of HNF4A, WFS1, DUSP9, FTO, and ZFAND6 2 genes are associated with Prediabetes Susceptibility and Inflammatory Markers in the Arab Population} describes the genetic association of specific genetic variants of the aforementioned genes with prediabetes in Saudi population.

Generally, it is a well-performed study, however, lacks clarity in presentation. Some of the major issues are grammatical errors, inappropriate punctuation, poor sentence construction, and spelling mistakes. The article must be thoroughly revised and English edited.

The title: please replace the Arab population with the Saudi population because the study does not include all Arabic population. Please change to Saudi throughout the manuscript.

Abstract: 

Please add more background about prediabetes and the selection of those genes under study.

Line 17 please replace Arab with Saudi.

The conclusion in lines 18-19 needs to be described more clearly.

Introduction: lines 36-38 please specify type 2 diabetes because the description and cited references are not related to type 1 diabetes mellitus which has different pathogenesis.

Please use T2DM instead of DM throughout the Manuscript, because the previously studied SNPs were related to T2DM.

Lines 40-41 need to be rephrased.

The introduction is brief and needs to address the results of previous genetic screening and should address some about the selected genes (HNF4A, WFS1, DUSP9, FTO, and ZFAND6 2).

Line 50-52 please clearly define prediabetes according to the international guidelines and mention blood glucose values. Add a sentence about impaired glucose tolerance and homeostasis.

Materials and Methods

Study Design and Population 

How the sample number was calculated? Why 332 prediabetic and 797 control subjects were selected?

Among the inclusion and exclusion criteria did you exclude morbidly obese patients?  

Line 117-118 (Details about genotype assignments 117 are described in our previous paper [19]).

Please rephrase you can write a brief description and add (has been performed according to our earlier described work…).

Line 94 lipid profile: please specify the measured parameters.

Line 95-100

Serological parameters are better replaced with pro-inflammatory cytokines.

Please specify the methods of the assay is it ELISA?

Please add the sensitivity of ELISA tests.

For assessment of prediabetes please add the values of blood glucose in mmol/l as you represented the results in mmol/l

Results:

Please write section 3.1 at the beginning of the results section.

Please write the full name before the appreciations throughout the whole manuscript for example Line 133 WHR. 

Please refrain from using significant/significantly multiple times in the same paragraph (especially significant p values are provided).

Use the journal style for (p-value) and make it consistent throughout the whole Manuscript.

Use the journal style of references [ ] and make it consistent in the whole manuscript (line 199).

Please add a description to the table (1), supplementary tables.

Please add fasting blood glucose in the table (1) instead of blood glucose.

Please comment on the inflammatory markers results in control and prediabetic subjects. 

Table (2) has been mentioned twice please correct the table numbers and add their description in the results section.

Line 164 (Table 2. Descriptive statistics of Anthropometrics according to select SNPs)

Please add a full table because the last column is not shown.

Please correct the number and the title of this table (according to the selected…) and it should be a table (3) and add the table description correctly (superscript B is not found). There is no description of this table in the text.

Discussion

Again the authors should specify T2DM throughout the discussion.

For line 198 GDM (gestational diabetes mellitus) please add the full name before the abbreviation at the beginning of each section.

Line 201 Please correct mature into maturity-onset diabetes (MODY 1).

Line 202-203 hence providing …. This sentence should be moved up after the references citing GDM because its current position is irrelevant to the previous sentence of MODY.

Conclusion:

258- 260

Please rephrase.

References.

Reference 27 please correct. 

Author Response

Reviewer 2

  1. Generally, it is a well-performed study, however, lacks clarity in presentation. Some of the major issues are grammatical errors, inappropriate punctuation, poor sentence construction, and spelling mistakes. The article must be thoroughly revised and English edited.

Response: We thank the reviewer for highlighting the issue; the necessary corrections have been made in the revised version of the manuscript

  1. The title: please replace the Arab population with the Saudi population because the study does not include all Arabic population. Please change to Saudi throughout the manuscript.

Response: We made it Saudi Arab instead of just Arab for appropriateness.

  1. Abstract: Please add more background about prediabetes and the selection of those genes under study. Line 17 please replace Arab with Saudi. The conclusion in lines 18-19 needs to be described more clearly.

Response: The conclusion was revised accordingly.

  1. Introduction: lines 36-38 please specify type 2 diabetes because the description and cited references are not related to type 1 diabetes mellitus which has different pathogenesis.

Response: It has been changed accordingly.

  1. Please use T2DM instead of DM throughout the manuscript, because the previously studied SNPs were related to T2DM.Lines 40-41 need to be rephrased.

Response: This point is well taken and T2DM was used entirely in the revised manuscript.

  1. The introduction is brief and needs to address the results of previous genetic screening and should address some about the selected genes (HNF4A, WFS1, DUSP9, FTO, and ZFAND6 2).

Response: The introduction was expanded substantially to provide additional insights in the genetic variants of interest.

  1. Line 50-52 please clearly define prediabetes according to the international guidelines and mention blood glucose values. Add a sentence about impaired glucose tolerance and homeostasis.

Response: The section was expanded accordingly.

  1. Materials and Methods. Study Design and Population. How the sample number was calculated? Why 332 prediabetic and 797 control subjects were selected?

Response: The subjects were randomly selected from the database where prediabetes status was not predetermined. We included in the revised results that from this cohort we were able to determine the prevalence of prediabetes which was 29%, all of whom were placed under the prediabetes group.

  1. Among the inclusion and exclusion criteria did you exclude morbidly obese patients?  

Response: We appreciate the reviewer’s comment. Indeed, morbidly obese subjects were excluded and this statement was inserted in the revised methods.

  1. Line 117-118 (Details about genotype assignments 117 are described in our previous paper [19]). Please rephrase you can write a brief description and add (has been performed according to our earlier described work…).

Response: It has been revised accordingly.

  1. Line 94 lipid profile: please specify the measured parameters.

Response: It has been specified.

  1. Line 95-100 Serological parameters are better replaced with pro-inflammatory cytokines.

Response: We appreciate this comment and the statement was revised accordingly.

  1. Please specify the methods of the assay is it ELISA? Please add the sensitivity of ELISA tests.

Response: We used multiplex immunoassays which is similar to ELISA but produces multiple signal measurements instead of single. Inter-and intra-assay sensitivity coefficients were already provided.

  1. For assessment of prediabetes please add the values of blood glucose in mmol/l as you represented the results in mmol/l

Response: This point is well taken and we replaced the units accordingly.

  1. Results: Please write section 3.1 at the beginning of the results section.

Response: Subsection 3.1 was included and the rest of the subsections in the results section were rearranged accordingly.

  1. Please write the full name before the appreciations throughout the whole manuscript for example Line 133 WHR.

Response: Abbreviations were now defined in first use.

  1. Please refrain from using significant/significantly multiple times in the same paragraph (especially significant p values are provided).

Response: It has been minimized

  1. Use the journal style for (p-value) and make it consistent throughout the whole manuscript.

Response: It was replaced accordingly.

  1. Use the journal style of references [ ] and make it consistent in the whole manuscript (line 199).

Response: It has been formatted accordingly.

  1. Please add a description to the table (1), supplementary tables.

Response: 

  1. Please add fasting blood glucose in the table (1) instead of blood glucose.

Response: It has been added.

  1. Please comment on the inflammatory markers results in control and prediabetic subjects. 

Response: Subsection 3.1 was expanded to describe the results on inflammatory markers.

  1. Table (2) has been mentioned twice please correct the table numbers and add their description in the results section.

Response: It has now been corrected.

  1. Line 164 (Table 2. Descriptive statistics of Anthropometrics according to select SNPs)

Response: It has now been corrected.

  1. Please add a full table because the last column is not shown.

Response: Tables were rearranged to portrait to facilitate better visualization.

  1. Please correct the number and the title of this table (according to the selected…) and it should be a table (3) and add the table description correctly (superscript B is not found). There is no description of this table in the text.

Response: It has now been corrected.

  1. Again the authors should specify T2DM throughout the discussion.

Response: T2DM was now used throughout the manuscript.

  1. For line 198 GDM (gestational diabetes mellitus) please add the full name before the abbreviation at the beginning of each section.

Response: GDM is now defined in first mention in the introduction.

  1. Line 201 Please correct mature into maturity-onset diabetes (MODY 1).

Response: Corrected.

  1. Line 202-203 hence providing …. This sentence should be moved up after the references citing GDM because its current position is irrelevant to the previous sentence of MODY.

Response: The statement was removed

  1. Conclusion: 258- 260 Please rephrase.

Response: The conclusion was rephrased accordingly.

  1. Reference 27 please correct. 

Round 2

Reviewer 1 Report

I want to express my gratitude for the opportunity to re-review the paper entitled: "Genetic Variants of HNF4A, WFS1, DUSP9, FTO, and ZFAND6 genes are associated with Prediabetes Susceptibility and Inflammatory Markers in the Arab Population" by Dalal N Binjawhar et al. Since the authors addressed all my concerns regarding the study design and the manuscript structure, I find it acceptable for publication in Genes.

Author Response

We sincerely the reviewer for appreciating the substantial revisions done.

Author Response

We thank the authors for reviewing the manuscript and response to revision. Now the manuscript has greatly improved. Here are some fewer comments.

  1. Still the introduction does not provide to the reader background importance of the selected gene in control of metabolism and carbohydrate metabolism hence the development of prediabetes.

Response: We thank the reviewer for this comment. Additional statements were added to give the needed background for the selected genes.

  1. Tables in the result sections and supplementary data need more description below each table.

Response: Footnotes of the main and supplementary tables were expanded accordingly.

  1. Table 3 line 208 (Table 3. Anthropometrics according to select SNPs) needs grammar correction it should be according to the selected SNPs.

Response: It has now been corrected.

  1. Table 4 line 221 (Table 4 Inflammatory markers according to select SNPs) needs grammar correction it should be according to the selected SNPs. Also, superscript A could not be detected in table. Please place superscripts correctly according to significant p value.

Response: The table title has now been corrected. Superscripts were placed accordingly with the p-value.